palaeontology, evolution

Cambrian, Chengjiang biota, micro-CT, *Naraoia spinosa*, arthropod, gnathobase

**Authors for correspondence:**
Xianguang Hou
e-mail: xghou@ynu.edu.cn
Yu Liu
e-mail: yu.liu@ynu.edu.cn

# Fine-scale appendage structure of the Cambrian trilobitomorph *Naraoia spinosa* and its ontogenetic and ecological implications

Dayou Zhai[1,2], Gregory D. Edgecombe[2,3], Andrew D. Bond[4], Huijuan Mai[1,2], Xianguang Hou[1,2] and Yu Liu[1,2]

[1]Yunnan Key Laboratory for Palaeobiology, Yunnan University, 2 North Cuihu Road, Kunming 650091, People's Republic of China
[2]MEC International Joint Laboratory for Palaeobiology and Palaeoenvironment, Yunnan University, 2 North Cuihu Road, Kunming 650091, People's Republic of China
[3]Department of Earth Sciences, The Natural History Museum, Cromwell Road, London SW75BD, UK
[4]Department of Earth Sciences, Royal Holloway University of London, Egham, Surrey TW200EX, UK

DZ, 0000-0001-6312-851X; GDE, 0000-0002-9591-8011

Trilobitomorphs are a species-rich Palaeozoic arthropod assemblage that unites trilobites with several other lineages that share similar appendage structure. Post-embryonic development of the exoskeleton is well documented for some trilobitomorphs, especially trilobites, but little is known of the ontogeny of their soft parts, limiting understanding of their autecology. Here, we document appendage structure of the Cambrian naraoiid trilobitomorph *Naraoia spinosa* by computed microtomography, resulting in three-dimensional reconstructions of appendages at both juvenile and adult stages. The adult has dense, strong spines on the protopods of post-antennal appendages, implying a predatory/scavenging behaviour. The absence of such gnathobasic structures, but instead tiny protopodal bristles and a number of endopodal setae, suggests a detritus-feeding strategy for the juvenile. Our data add strong morphological evidence for ecological niche shifting by Cambrian arthropods during their life cycles. A conserved number of appendages across the sampled developmental stages demonstrates that *Naraoia* ceased budding off new appendages by the mid-juvenile stage.

## 1. Introduction

Naraoiids are an abundant group of non-biomineralized trilobitomorph arthropods, their exoskeleton organized as two lightly sclerotized shields jointed by a single articulation between the head and trunk [1–4]. First discovered in the Burgess Shale (Miaolingian Series, British Columbia), naraoiids range from Cambrian Series 2 to the Přidolian (Silurian) [5–7]. They are of broader interest for studies of early arthropods because of the exquisite detail in which their appendages are known. This information is mostly derived from specimens from the Chengjiang biota of south China (Cambrian Series 2) [2,3,8] and the Burgess Shale [1,4,9], allowing for the precise structure of the two limb rami, the morphology of the proximal part of the appendage, and the attachment of the appendages to the body to be resolved to an almost unrivaled degree within the trilobitomorphs.

Despite their non-biomineralized exoskeleton, naraoiids were classified as members of Trilobita, which are otherwise robustly calcified [1,10]. Hou & Bergström [8] instead proposed that trilobites and naraoiids are not each other's sister group within an assemblage of mostly non-biomineralized Palaeozoic arthropods named Artiopoda. The Artiopoda concept was underpinned by the serial similarity in the form of all post-antennal biramous limbs, and a distinctively trilobite-type appendage and monophyly in phylogenetic analyses have supported

Artiopoda [11,12]. Several subsequent studies have recovered separate sister groups for both trilobites and naraoiids [13–15], though all of these taxa are consistently recovered within a subgroup of Artiopoda that corresponds to a monophyletic formulation of Trilobitomorpha (see fig. 6 of [11]).

Structure of the appendages has played a role in inferring the life habits of naraoiids. Whittington [1] and Caron & Jackson [16] considered naraoiids to be benthic from their interpretation of feeding strategies and relative abundance. Chen et al. [2] interpreted N. spinosa as being a benthic deposit feeder in view of a sediment-filled gut, laterally deflected antennae and preservational posture, though these were contested by subsequent studies which favoured a predatory/scavenging mode of life in light of digestive morphology [3,17]. A nekto-benthic mode of life for naraoiids has alternatively been proposed, viewing the appendages of *Naraoia compacta* as specialized for swimming [9]. However, no specialized swimming appendages have been discovered within other naraoiid species or genera.

In this study, we undertake high-resolution micro-computed tomography (CT) study of both the juvenile and adult specimens of *N. spinosa*. Micro-CT reconstructions reveal a full set of three-dimensional appendages with micrometre-scale details, as well as marked changes of appendicular structures between ontogenetic stages. These observations have allowed for a revised interpretation of naraoiid feeding and locomotory strategies.

## 2. Material and methods

### (a) Materials

The specimens (YKLP 11408, 11409, 13941) were collected from the Haikou area of Kunming, China and are housed at Yunnan Key Laboratory for Palaeobiology, Yunnan University. They are preserved in the yellowish mudstones in the Yu'anshan Member of the Chiungchussu Formation, Cambrian Stage 3. The specimens are partially preserved as pyrite and/or iron oxide, as is common for Chengjiang fossils [18], which confers a density difference with the matrix that is conducive to X-ray microtomographic imaging.

### (b) Fossil imaging

Microscopic photos were taken with a Leica DFC500 CCD camera attached to a Leica M205C stereomicroscope and a Leica M205FA fluorescence microscope, respectively. In order to image the buried structures, X-ray computed microtomography was applied. Specimens YKLP 11408 and YKLP 11409 (figures 1 and 2; electronic supplementary material, figure S1) were scanned with a GE Phoenix Nanotom m scanner. Scanning energy was set at 110 kV/100 µA and imaging pixel size ranged from 16.2 to 4.0 µm, with coarse resolution for large-field scanning and fine resolution for small-field scanning. Specimen YKLP 13941 (electronic supplementary material, figure S2) was scanned with a Zeiss Xradia 520 Versa X-ray Microscope, with scanning energy of 70 kV/86 µA and imaging pixel size of 9.0 µm. The resulting data, in the form of a suite of TIFF images (one to a few thousand in number), were processed with the software DRISHTI (v. 2.4 and v. 2.6.4) for three-dimensional observations of the fossils.

## 3. External morphology

The specimens documented herein are identified as *Naraoia spinosa* Zhang & Hou (1985), one of two common naraoiids in the Chengjiang biota, based on their general outlines, appendage morphologies and the characteristic gut diverticula on the head shield (figures 1 and 2; electronic supplementary material, figures S1 and S2). The occurrence of marginal spines on the dorsal exoskeleton further indicates affinity to 'morph A' [3]. We consider YKLP 11408 (length 11.37 mm, figure 1) as a mid-stage juvenile based on length-frequency data for this morph (fig. 17 in [3]). We refer to YKLP 11409 (length 30.74 mm, figure 2) as an adult based on its size, trunk shield that is significantly longer than the head shield, tiny spines along the posterior margin of the trunk shield (green arrowheads in figure 2a; cf. [2,3]), and appendage structures being more developed compared to those of the juvenile (see below), although the largest known specimens of the species have a body length of 5 cm [19].

The exoskeleton consists of a semicircular head shield and an elongate trunk shield that overlap each other only slightly (figures 1 and 2). Marginal spines are present on the trunk shield. In the adult, the last pair of these spines is much larger than the preceding ones (figure 2a).

The hypostomal complex consists of a triangular, anteriorly pointing plate and two subtriangular (though evidently broken), ventrally projecting plates (ap and vp, figures 1e, 2j and 3). The anterior plate of the juvenile is longer (length 1.44 mm) and is situated more anteriorly than that of the adult (length 1.20 mm). The multi-articulated antennae are inserted between the anterior plate and the ventral plates.

Numbers of appendages (i.e. 20 pairs) are identical in the juvenile (figure 1a) and the adult (figure 2a). However, detailed morphology and the pattern of segmental differentiation of these appendages vary between them. In the juvenile, the post-antennal appendages are represented by two types based on protopodal armature, dividing the body tagmosis into two sections. In appendages 2–6, the enlarged, quadrate protopod carries two rows of variably tiny bristles on the inner edge and scattered bristles on the lateral surface (figures 1b,f and 3). Small bristles also occur on the prominent tooth-like endite of the first endopodal podomere, where they are less clearly patterned (figure 1f). A first endopodal podomere bearing a prominent endite is well known from other naraoiid species [1,3,8,9], but the detailed three-dimensional morphology of both the protopod and first endopodal podomere is clarified here. A similar endite, albeit being smaller, is present on the second endopodal podomere but not on subsequent ones. As observed previously (fig. 4m in [11]) the endopod is composed of six podomeres and a terminal claw. The distal endopodal podomeres are slender, each with a pair of tiny sub-apical endites that bear single setae (figures 1b–d,g and 3). The terminal claw is pointed and triangular. The exopod flap is connected to the protopod along a strong shaft, the morphology of which is clearly revealed (figure 1b,c). Appendage 6 lies under the trunk based on three-dimensional rotation of the specimen to trace the head-trunk boundary onto the ventral side, on which it is visible as a transverse line.

The more posterior trunk appendages differ from appendages 2–6 in several aspects. Numerous segments depict three or, mostly, four sub-equal lobes (figure 1a,h, green arrowheads) that are more likely folds of arthrodial membrane at the attachment of the protopod to the body than representing lobate endites of the protopod itself. These lobes (the 'cormus' in fig. 45B of [8]) have a high fossilization potential in naraoiids, being documented in many specimens of *Misszhouia longicaudata* in particular [2,8], and their morphology and position are similar in our material and *M. longicaudata*. Whether these structures are present in the head appendages of YKLP 11408 is unknown, because the central part of the head shield of this specimen is

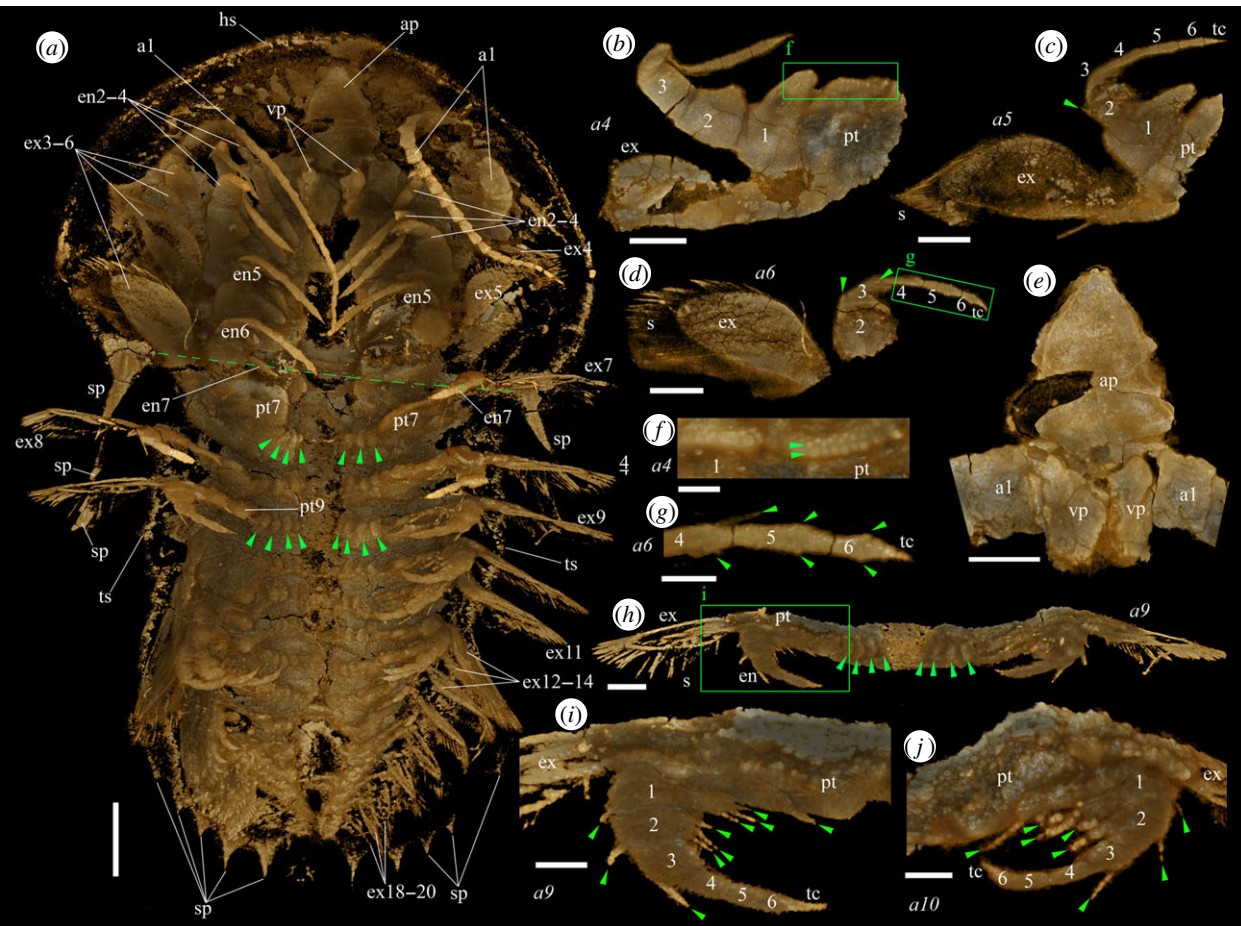

**Figure 1.** Micro-CT images of a mid-stage juvenile specimen (YKLP 11408) of *Naraoia spinosa* Zhang & Hou (1985) from the Chengjiang Lagerstätte (Cambrian Series 2, Stage 3). (*a*) Ventral view of the animal. Green arrowheads indicate folds of arthrodial membrane proximal to the protopods of seventh and ninth appendages, respectively. Green dashed line shows position of posterior margin of head shield. (*b*) Right fourth appendage, posterior view. (*c*) Right fifth appendage, posterior view. (*d*) Right sixth appendage, posterior view. (*e*) Hypostomal complex and basal parts of first appendages (antennae), ventral view. (*f*) Enlargement of the area in the green rectangle in (*b*) but in ventral view. Green arrowheads indicate the two rows of bristles on protopod. (*g*) Enlargement of the area in the green rectangle of (*d*). Green arrowheads indicate endites at the sub-apical part of the endopodal podomeres. Note the seta on the endite of the fourth podomere. (*h*) Ninth appendages, ventral view. Green arrowheads indicate folds of arthrodial membrane proximal to the protopod (see also (*a*)). (*i*) Enlargement of the area in the green rectangle of (*h*). Green arrowheads indicate endopodal setae. (*j*) Part of left tenth appendage, oblique-ventral view. Green arrowheads indicate endopodal setae. Scale bars, 1 mm for (*a*), 500 μm for (*b–e*, *h*) and 200 μm for (*f*, *g*, *i*, *j*). ap, anterior plate of the hypostomal complex; a*n*, the *n*th appendage; en, endopod; en*x*, endopod of *x*th appendage; ex*n*, exopod of *n*th appendage; hs, head shield; pt*n*, protopod of *n*th appendage; s, seta; sp, spine; tc, terminal claw; ts, trunk shield; vp, ventral plate of the hypostomal complex. Numerals 1–6 indicate endopodal podomeres. (Online version in colour.)

missing (figure 1*a*), although their similar expression in head and trunk segments in *M. longicaudata* (fig. 9A in [2]) suggests they are likely to be present in the head of *N. spinosa* as well. The protopod of the posterior trunk appendages carries a bare, blunt lobe (figure 1*i*,*j*). Its size may have been taphonomically reduced in the ventrally compressed appendages 8–20 compared to the seventh appendage, where it is generally anteriorly positioned, with the size of protopod being similar to those of appendages 2–6 (figure 1*a*). Well-developed setae are present on the first three endopodal podomeres but not on the distal podomeres (figure 1*i*,*j*), and the endopodal podomeres generally become progressively more slender distally, rather than having the distal four podomeres uniformly slender as in appendages 2–6. The exopod shaft is relatively shorter than in appendages 2–6 and the flap is more slender, but the two are likewise separated by a distinct articulation. The distal paddle bears spiniform setae with rounded cross sections.

In comparison to the juvenile, the post-antennal appendages of the adult have generally spinose protopods, the spines being more pronounced on the anterior appendages (figures 2 and 3). In appendages 2 and 3 (figure 2*b*,*e*,*f*), the

spine assemblage consists of one or two larger and about 30 more slender spines. The larger spines usually curve slightly proximally. In the middle appendages (e.g. appendage 9, figure 2*c*,*g*), spines are crudely arranged in three or four rows, mostly of similar size but slightly longer on the distal part of the protopod, and slightly curved towards the proximal of the appendage. In the more posterior appendages (e.g. appendages 14 and 15, figure 2*h*,*i*), these spines generally become shorter and blunter, their morphology resembling the protopodal bristles of the juvenile (figure 1*f*), although much larger. The endopod of the adult (as in the juvenile) consists of six podomeres and a pointed terminal claw. The endopod (e.g. appendage 10, figure 2*d*) has a robust seta articulated ventrodistally on several articles (figure 2*d*). Morphology of the exopods is generally similar to those of the juvenile, being flap-like and densely fringed with setae (figure 2*a*,*d*). However, a shaft is indistinctly defined, and the exopod flap itself is relatively larger, especially on the cephalic appendages.

Gnathobasic spines are also observed on several trunk segments in a late-stage juvenile (YKLP 13941; electronic supplementary material, figure S2, green arrowheads), which has

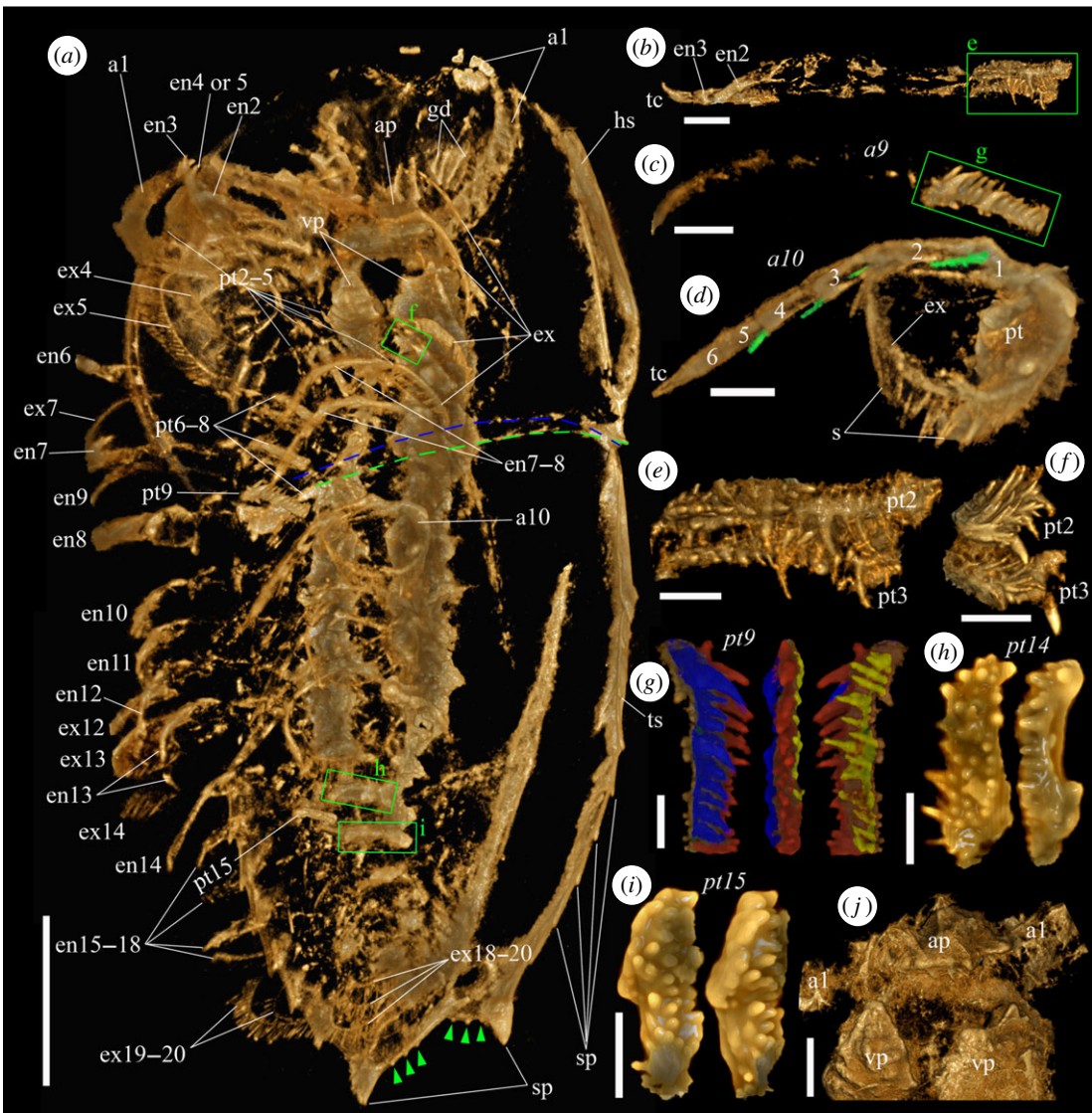

**Figure 2.** Micro-CT images of an adult specimen (YKLP 11409) of *Naraoia spinosa* from the Chengjiang Lagerstätte. (*a*) Ventral view of the animal. Green dashed line marks the position of the posterior margin of the head shield, blue dashed line for the anterior margin of the trunk shield. Green arrowheads indicate tiny spines on the posterior margin of the trunk shield. (*b*) Right second and third appendages, exopods not preserved. (*c*) Right ninth appendage, exopod not preserved. (*d*) Left tenth appendage. Green coloration highlights apical setae on endopodal podomeres 1–4. (*e*) Protopods of right second and third appendages, interior view (green rectangle in (*b*)). (*f*) Protopods of left second and third appendages, interior view (one of the green rectangles in (*a*), viewing from the back side). (*g*) Protopod of right ninth appendage (green rectangles in (*c*)) in posterior, interior and anterior views. Blue, red and yellow colorations highlight different rows of spines. (*h*) Protopod of left 14th appendage in ventral and posteroventral views. (*i*) Protopod of left 15th appendage in ventral and anteroventral views. (*j*) Hypostomal complex and basal parts of first appendages (antennae), ventral view. Scale bars, 5 mm for (*a*), 1 mm for (*b–d*, *j*) and 500 μm for (*e–i*). Abbreviations as in figure 1. Numerals 1–6 indicate endopodal podomeres. (Online version in colour.)

a body length of 21 mm and has protopodal spines that are less pronounced than those in the adult (figure 2). Due to limits of preservation, however, the arrangement of these spines on the protopod and their presence in the head appendages are not clearly defined in this specimen.

## 4. Discussion

Our study presents unprecedently clear, three-dimensional morphologies of the entire appendage assemblage of the Cambrian arthropod *Naraoia spinosa*. We acknowledge that micro-CT can sometimes be biased by differential pyritisation of structures, as seen by the variable amount of detail preserved along the length of the body in single specimens. Accordingly, the effort has been taken to integrate our data with light

photographic observations on *Naraoia spinosa* in previous studies [3], as well as other species of *Naraoia* and naraoiids more broadly. We infer, for example, that the tomographic data are often exposing only the marginal part of the protopods in the scanned adults (where spines are concentrated) because protopods observed in fully exposed specimens are prominently extended (as reconstructed in fig. 30 of [3]). As for specimen YKLP 11408, in view of its generally complete appearance and rich seta-scale details (figure 1), we believe micro-CT captures most of its external morphology.

It is especially noticeable that the protopods are modified into various forms along the body axis in different ontogenetic stages, presumably for food-processing purposes. The endopods and exopods are, by contrast, more conserved between body regions. In the adult, the densely distributed, strong protopodal spines arranged in multiple rows with long and short

Proc. R. Soc. B 286: 20192371

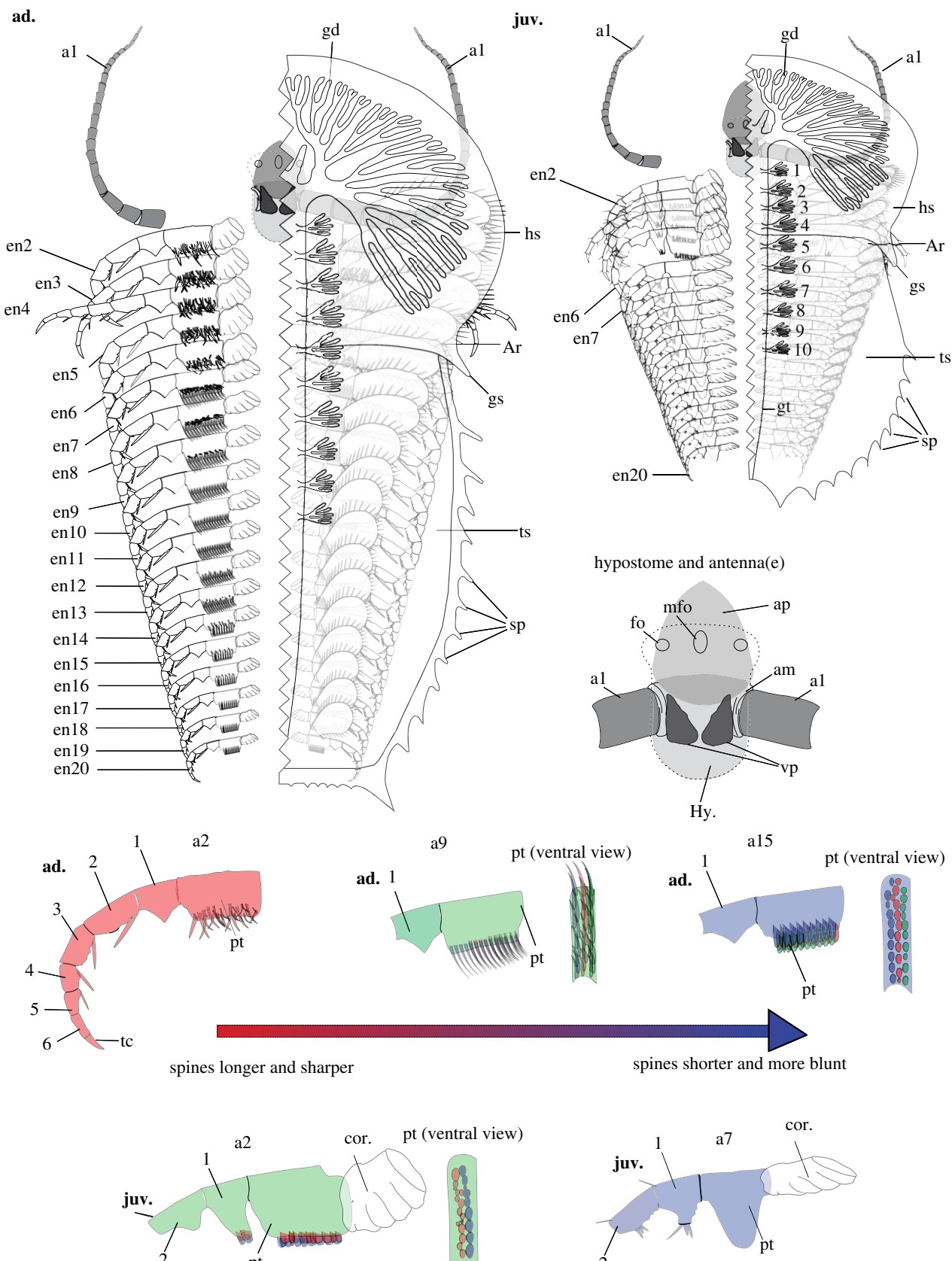

**Figure 3.** (*Caption overleaf.*)

spines (figure 2) resemble those of extant arthropods such as horseshoe crabs, implying the capabilities of tearing hard food items, and therefore predatory and/or scavenging behaviour [20]. This supports the conclusion drawn from comparison of the digestive system of *N. spinosa* with Recent arthropods [17]. As well, the adult of *N. spinosa* depicts a morphological

trend of shorter but more robust spines on the protopods of posterior appendages, consistent with feeding behaviour in horseshoe crabs and durophagous Cambrian arthropods such as *Sidneyia*, in which the posterior appendages break up prey that is passed forward to the more delicately spinose anterior appendages [21]. No shelly fragments are known

**Figure 3.** (*overleaf.*) Line drawings showing differentiation of the appendages of *Naraoia spinosa* and the difference between juvenile and adult. *Left-hand upper:* a detailed line drawing of *N. spinosa*, of early-adult stage of growth (ad.), based on YKLP 11409. *Right-hand upper:* a detailed line drawing of *N. spinosa*, of mid-juvenile stage of growth (juv.), based on YKLP 11408. The left-hand side of both the adult and juvenile specimens has been removed to better reveal the morphologies and arrangement of the appendages. The exopods on the left-hand side of both specimens, as well as the proximal-most portion of the left antenna, have also been removed to clearly display the *in situ* morphologies of the protopods. Morphology of posterior shield and gut morphology, within both the adult and the juvenile, based on fig. 20 of [3]. *Right-hand centre:* an enlarged line drawing of the hypostomal complex of *N. spinosa* based on both YKLP 11408 and 11409. General hypostome outline and positioning of frontal organs based on fig. 21 of [3]. *Bottom of figure:* detailed line drawings exhibiting the protopod morphologies of both mid juvenile (juv.) and early adult (ad.) specimens of *N. spinosa* based on YKLP 11408 and 11409. No. 1–10, numbered gut diverticula (main specimens) and numbered endopodal podomeres (dissected structures); ad., morphologies pertaining to adult specimens of *N. spinosa*; am, inferred presence of arthrodial membrane; a*n*, the *n*th appendage; ap, anterior plate of the hypostomal complex; Ar, shield articulation; en*x*, the endopod of *x*th appendage; fo, the lateral frontal organs of the hypostome complex; gd, gut diverticula; gs, genal spine; gt, gut tract; hs, head shield; Hy., hypostome; juv., morphologies pertaining to juvenile specimens of *N. spinosa*; mfo, medial frontal organ of the hypostomal complex; pt, protopod; sp, spine; tc, terminal claw; ts, trunk shield; vp, ventral plate of the hypostomal complex. Numerals 1–6 indicate endopodal podomeres. (Online version in colour.)

from the gut of our scanned specimens or other material of *N. spinosa*, making durophagy less confident than is the case for *Sidneyia*, for example [22,23]. In the juvenile, the small bristles on the protopod of the head appendages (figure 1*b,f*) probably functioned for grinding smaller, softer food particles. The less-developed trunk appendages with (apparently) a bare protopod lobe and setulose endopods (figure 2*i,j*) may have been used to collect and transfer food particles, such as organic detritus and small plankton, to the head appendages for further processing. We suggest that the juvenile of *N. spinosa* had a different feeding strategy from the adult, relying on smaller prey or even being a plankton/detritus feeder. Such a finding echoes that by [24], who proposed a diet difference between the juvenile and adult of the Cambrian 'great appendage' arthropod *Leanchoilia illecebrosa* based on ontogenetic differences in the deutocerebral great appendages. Moreover, we infer that the juvenile and adult of *N. spinosa* may also differ in their locomotory behaviour and respiratory modes. As shown in figures 1–3, exopods of the adult, especially from the posterior part of the body, are generally larger, and more densely fringed with setae compared with those of the juvenile. This may reflect differences in the animals' ability to swim and obtain dissolved oxygen from the ambient water.

The pattern of increasing spinosity of the gnathobases through ontogeny is also similar to many extant marine arthropods, as seen, for example, in *Limulus* [25]. This suggests that this trend may be a plesiomorphic character shared by other artiopodans, but few comparable data are available to test its distribution. In the case of agnostids, a clade that has long been the subject of debate as to whether or not they are allied to trilobites (reviewed by Moysiuk & Caron [26]), comparison of juveniles [27] and adults suggests limited ontogenetic change [26], although comparison is only possible between different genera, and the adult morphology is known in much coarser detail than is available for *N. spinosa*.

Artiopodans are generally thought to have limited differentiation of the post-antennal appendages. Our data for *N. spinosa* depict marked differentiation of the protopods along the body axis throughout ontogeny. In Trilobita, the spinosity, size of endites and orientation of the protopod have likewise been noted to vary along the anteroposterior axis in *Eoredlichia intermedia* and *Triarthrus eatoni* (reviewed in [28]), whereas *Redlichia rex* depicts differences in relative size of the lobes of the exopod between anterior and posterior appendages [28]. The recent finding that the first two post-antennal appendages of the xandarellid artiopodan *Sinoburius lunaris* are markedly differentiated from other cephalic and trunk

appendages with respect to exopod morphology [29] suggests that appendages may be more tagmatized in this clade than previously known.

Our study also provides important information on the developmental mode of *N. spinosa*. It is found that the juvenile, which is about one-third the length of the studied adult (and just over one-fifth the length of the largest known specimens of the species), possesses the same number of appendages as the adult. This indicates that *Naraoia* attained the full appendage assemblage at an early ontogenetic stage, although the size, shape, segmentation pattern and chaetotaxy structures of various parts of the appendages continue to change afterwards. Such a pattern resembles that of *Leanchoilia illecebrosa* [30], for which a 2-mm-long juvenile already had nearly the same number of appendages as the adult (lengths up to 5 cm [19]). It is not clear, however, whether or not *N. spinosa* is strictly epimorphic, having already completed its segmentation in the embryo, but it contrasts with the overwhelming majority of trilobites, which have anamorphic development, adding body segments for a variable (and often lengthy) duration during postembryonic development before reaching the epimorphic phase. The 'segmentation schedule' of trilobites [31] describes differences in timing between segment addition at the posterior growth zone and articulation between segmental sclerites. Unlike trilobites (and most other trilobitomorphs), naraoiids have a fused trunk shield and thus their segmentation schedule does not involve the process of articulation. Post-embryonic addition of segments (a hemianamorphic phase), which might have occurred at stages earlier than the larva studied here, would involve the trunk shield becoming relatively longer, assuming a typical pre-telson budding zone.

Data accessibility. The computed tomography data are available on Dryad Digital Repository: https://dx.doi.org/10.5061/dryad.jdfn2z372.

Authors' contributions. X.H. collected the fossils. D.Z. and Y.L. designed the research and processed micro-CT data with input from H.M. D.Z. and A.D.B. prepared the figures with input from co-authors. D.Z., G.D.E., Y.L. and A.D.B. wrote the first draft of the manuscript. All authors discussed and approved the manuscript.

Competing interests. We declare we have no competing interests.

Funding. This study is supported by NSFC grant nos. 41861134032, 41472153, Yunnan Provincial grant nos. 2018FA025, 2018IA073, 2015HA021 and 2015HC029, and 1000 Talent Plan of China (Youth Program).

Acknowledgement. We thank Hong Chen, Ting Zhao and Maoyin Zhang for photographing the specimens, and Shuyan Hou and Shaogang Zang for scanning the fossils.

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
