## [Reviewer comments · Proceedings of the Royal Society B: Biological Sciences]

Review History

RSPB-2019-1721.R0 (Original submission)

Review form: Reviewer 1 (Omar Rafael Regalado Fernandez)

Recommendation

Accept with minor revision (please list in comments)

Scientific importance: Is the manuscript an original and important contribution to its field?

Good

General interest: Is the paper of sufficient general interest?

Acceptable

Quality of the paper: Is the overall quality of the paper suitable?

Excellent

Is the length of the paper justified?

No

Should the paper be seen by a specialist statistical reviewer?

No

Do you have any concerns about statistical analyses in this paper? If so, please specify them explicitly in your report.

No

It is a condition of publication that authors make their supporting data, code and materials available - either as supplementary material or hosted in an external repository. Please rate, if applicable, the supporting data on the following criteria.

Is it accessible?

Yes

Is it clear?

Yes

Is it adequate?

Yes

Do you have any ethical concerns with this paper?

No

Comments to the Author

The anatomical insights gained by computed microtomography are an excellent opportunity to produce new phylogenetic hypotheses. This paper needs a more thorough comparison with other trilobitormorphs pointing out specific trends (ecological or ontogenetic) found in *Naraoia* and other artiopodans. A phylogenetic analysis adding this new information into an existing matrix could allow making more sound inferences about the anatomy of other closely related arthropods, such as xandarellids. I would suggest complementing the discussion in this paper by exploring the evolutionary affinities of *Naraoia* by expanding the matrix published in Moysiuk and Caron (2019).

Moysiuk J, Caron J-B. 2019. Burgess Shale fossils shed light on the agnostid problem. *Proc. R. Soc. B* 286:20182314

Review form: Reviewer 2

Recommendation

Accept with minor revision (please list in comments)

Scientific importance: Is the manuscript an original and important contribution to its field?

Excellent

General interest: Is the paper of sufficient general interest?

Excellent

Quality of the paper: Is the overall quality of the paper suitable?

Excellent

Is the length of the paper justified?

Yes

Should the paper be seen by a specialist statistical reviewer?

No

Do you have any concerns about statistical analyses in this paper? If so, please specify them explicitly in your report.

No

It is a condition of publication that authors make their supporting data, code and materials available - either as supplementary material or hosted in an external repository. Please rate, if applicable, the supporting data on the following criteria.

Is it accessible?

Yes

Is it clear?

Yes

Is it adequate?

Yes

Do you have any ethical concerns with this paper?

No

Comments to the Author

General comments

This is an excellent paper that describes the exquisitely preserved 3D morphology of juvenile and adult specimens of the Cambrian arthropod *Naraoia spinosa* from the Chengjiang Biota of China. It provides much needed new data on the morphology of this iconic group, and presents important novel information about the development and feeding modes of one of the oldest arthropods in the fossil record. Overall, the descriptions are accurate, the discussion is well written and argued, and the illustrations are of a high standard and show the necessary details. I think the manuscript could be published after very minor revision, provided the authors consider the points below:

1. It might be worth mentioning in the 'Materials' section that the specimens are partially preserved as pyrite and/or iron oxide. This preservation mode, with structures such as the appendages being replicated by a much denser material than the surrounding matrix, is obviously why such specimens are conducive for micro-CT scanning and 3D imaging. Perhaps also citing a reference or two that discusses this preservation mode in Chengjiang fossils might be useful?
2. I think there could be a bit more discussion on the functional differentiation of gnathobase morphology along the A-P axis of the adult, particularly in the context of other Cambrian and modern taxa. It is clear from the figures (especially Fig. 3) that the gnathobasic spines become more robust on the more posterior appendage pairs. This is a morphological trend seen in adult specimens of durophagous arthropods such as *Limulus polyphemus* and the Cambrian taxon

Sidneyia inexpectans – see a recent discussion by Bicknell et al. (2018, *Arthropod Structure & Development*, 47: 12-24). This arrangement indicates that the posterior appendages are used for crushing harder food items before passing them anteriorly towards the mouth, where they are manipulated by the more delicate spines on the anteriormost pairs of gnathobases.

3. This may be beyond the scope of the present study, but have the authors carefully examined specimens of *N. spinosa* in their collections that have digestive (gut) tracts preserved to see if there are shelly fragments contained within? Cololites provide ‘smoking gun’ evidence that the animal was indeed capable of durophagy (see the discussion by Bicknell & Paterson 2018, *Biological Reviews*, 93: 754-784), thus further supporting the claims made here. I should be clear that the manuscript does not need to provide this evidence, as the arguments about the feeding modes of juvenile and adult *N. spinosa* is already well supported by the new morphological information, but it would be great to see new data on Cambrian cololites if it is available. Just a suggestion.

Decision letter (RSPB-2019-1721.R0)

12-Sep-2019

Dear Dr Zhai:

I am writing to inform you that your manuscript RSPB-2019-1721 entitled "Fine-scale appendage structure of the Cambrian trilobitomorph *Naraoia spinosa* and its ontogenetic and ecological implications" has, in its current form, been rejected for publication in *Proceedings B*.

This action has been taken on the advice of referees, who have recommended that substantial revisions are necessary. With this in mind we would be happy to consider a resubmission, provided the comments of the referees are fully addressed. However please note that this is not a provisional acceptance.

Indeed, while the reviewers' comments are brief and positive, it is clear that the MS needs to be broader in scope (requiring substantial changes-- such as inclusion of plenty more taxa for comparison, ideally outside arachnomorphs) and more phylogenetic analysis, to satisfy the reviewers and achieve the broad scope and interest level suitable to *Proc B*.

- 1) A ‘response to referees’ document including details of how you have responded to the comments, and the adjustments you have made.
- 2) A clean copy of the manuscript and one with 'tracked changes' indicating your 'response to referees' comments document.
- 3) Line numbers in your main document.

In your revision process, please take a second look at how open your science is; our policy is that all data involved with the study should be made openly accessible-- see: <https://royalsociety.org/journals/ethics-policies/data-sharing-mining/>
Insufficient sharing of data can delay or even cause rejection of a paper.

Sincerely,
Professor John Hutchinson
mailto: proceedingsb@royalsociety.org

Associate Editor

Comments to Author:

Thank you for submitting your manuscript to Proceedings B. The reviewers have come back with rather contrasting reviews. Whilst Reviewer 2 is very happy with the manuscript and has very minor suggestions, Reviewer 1 would like to see considerable modifications. In order for the manuscript to appeal to the broad readership of PRSB, Reviewer 1 has suggested that the ontogeny of *Naraoia* be better framed within a broader comparison of the appendage anatomy of other taxa (e.g. *Tegopelte*, *Xandarella*, and trilobites such as agnostids). Additionally, Reviewer 1 has suggested the authors revisit the phylogenetic affinities of *Naraoia* in light of the recent phylogenetic analysis of Moysiuk and Caron (2019).

Reviewer(s)' Comments to Author:

Referee: 1

Comments to the Author(s)

The anatomical insights gained by computed microtomography are an excellent opportunity to produce new phylogenetic hypotheses. This paper needs a more thorough comparison with other trilobitomorphs pointing out specific trends (ecological or ontogenetic) found in *Naraoia* and other arthropods. A phylogenetic analysis adding this new information into an existing matrix could allow making more sound inferences about the anatomy of other closely related arthropods, such as xandarellids. I would suggest complementing the discussion in this paper by exploring the evolutionary affinities of *Naraoia* by expanding the matrix published in Moysiuk and Caron (2019).

Moysiuk J, Caron J-B. 2019. Burgess Shale fossils shed light on the agnostid problem. *Proc. R. Soc. B* 286:20182314

Referee: 2

Comments to the Author(s)

General comments

This is an excellent paper that describes the exquisitely preserved 3D morphology of juvenile and adult specimens of the Cambrian arthropod *Naraoia spinosa* from the Chengjiang Biota of China.

It provides much needed new data on the morphology of this iconic group, and presents important novel information about the development and feeding modes of one of the oldest arthropods in the fossil record. Overall, the descriptions are accurate, the discussion is well written and argued, and the illustrations are of a high standard and show the necessary details. I think the manuscript could be published after very minor revision, provided the authors consider the points below:

1. It might be worth mentioning in the 'Materials' section that the specimens are partially preserved as pyrite and/or iron oxide. This preservation mode, with structures such as the appendages being replicated by a much denser material than the surrounding matrix, is obviously why such specimens are conducive for micro-CT scanning and 3D imaging. Perhaps also citing a reference or two that discusses this preservation mode in Chengjiang fossils might be useful?
2. I think there could be a bit more discussion on the functional differentiation of gnathobase morphology along the A-P axis of the adult, particularly in the context of other Cambrian and modern taxa. It is clear from the figures (especially Fig. 3) that the gnathobasic spines become more robust on the more posterior appendage pairs. This is a morphological trend seen in adult specimens of durophagous arthropods such as *Limulus polyphemus* and the Cambrian taxon *Sidneyia inexpectans* – see a recent discussion by Bicknell et al. (2018, *Arthropod Structure & Development*, 47: 12-24). This arrangement indicates that the posterior appendages are used for crushing harder food items before passing them anteriorly towards the mouth, where they are manipulated by the more delicate spines on the anteriormost pairs of gnathobases.
3. This may be beyond the scope of the present study, but have the authors carefully examined specimens of *N. spinosa* in their collections that have digestive (gut) tracts preserved to see if there are shelly fragments contained within? Cololites provide 'smoking gun' evidence that the animal was indeed capable of durophagy (see the discussion by Bicknell & Paterson 2018, *Biological Reviews*, 93: 754-784), thus further supporting the claims made here. I should be clear that the manuscript does not need to provide this evidence, as the arguments about the feeding modes of juvenile and adult *N. spinosa* is already well supported by the new morphological information, but it would be great to see new data on Cambrian cololites if it is available. Just a suggestion.

Author's Response to Decision Letter for (RSPB-2019-1721.R0)

See Appendix A.

RSPB-2019-2371.R0

Review form: Reviewer 1 (Dr Omar Rafael Regalado Fernandez)

Recommendation

Accept as is

Scientific importance: Is the manuscript an original and important contribution to its field?
Excellent

General interest: Is the paper of sufficient general interest?
Good

Quality of the paper: Is the overall quality of the paper suitable?
Good

Is the length of the paper justified?
Yes

Should the paper be seen by a specialist statistical reviewer?
No

Do you have any concerns about statistical analyses in this paper? If so, please specify them explicitly in your report.
No

It is a condition of publication that authors make their supporting data, code and materials available - either as supplementary material or hosted in an external repository. Please rate, if applicable, the supporting data on the following criteria.

Is it accessible?
Yes

Is it clear?
Yes

Is it adequate?
Yes

Do you have any ethical concerns with this paper?
No

Comments to the Author

This new resubmission is more informative than the previous one and the comparative anatomy with other arthropods is insightful.

Review form: Reviewer 2

Recommendation
Accept as is

Scientific importance: Is the manuscript an original and important contribution to its field?
Excellent

General interest: Is the paper of sufficient general interest?
Excellent

Quality of the paper: Is the overall quality of the paper suitable?

Excellent

Is the length of the paper justified?

Yes

Should the paper be seen by a specialist statistical reviewer?

No

Do you have any concerns about statistical analyses in this paper? If so, please specify them explicitly in your report.

No

It is a condition of publication that authors make their supporting data, code and materials available - either as supplementary material or hosted in an external repository. Please rate, if applicable, the supporting data on the following criteria.

Is it accessible?

Yes

Is it clear?

Yes

Is it adequate?

Yes

Do you have any ethical concerns with this paper?

No

Comments to the Author

I am satisfied with the authors' responses to the referee comments, as well as the revised manuscript. I think it is now ready for publication, pending two very minor points to be addressed:

1. The year for the Bicknell & Paterson reference should be 2018 (not 2017)
2. In the new section on appendage tagmatization (lines 229-234), a reference could be made to another recent study (Holmes et al. 2019, Jour. Systematic Palaeo) that suggests some trilobites show possible anterior-posterior differentiation of the biramous limbs - in this particular case, the exopods of *Redlichia rex* (see their fig. 20), though Holmes et al. (2019) note that similar A-P trends occur in the appendages of other trilobite taxa such as *Eoredlichia intermediata* and *Triarthrus eatoni*. So it would seem A-P differentiation of the appendages in artiopodans could be quite widespread across the clade, but expressed in different ways. Something to consider.

Decision letter (RSPB-2019-2371.R0)

24-Oct-2019

Dear Dr Zhai

I am pleased to inform you that your manuscript RSPB-2019-2371 entitled "Fine-scale appendage

structure of the Cambrian trilobitomorph *Naraoia spinosa* and its ontogenetic and ecological implications" has been accepted for publication in Proceedings B.

The referee(s) have recommended publication, but also suggest some minor revisions to your manuscript. They also had hoped for a phylogenetic analysis and you might still provide that but they did not push for it as a mandatory inclusion. Therefore, I invite you to respond to the referee(s)' comments and revise your manuscript. Because the schedule for publication is very tight, it is a condition of publication that you submit the revised version of your manuscript within 7 days. If you do not think you will be able to meet this date please let us know.

[http://datadryad.org/submit?journalID=RSPB&manu=\(Document not available\)](http://datadryad.org/submit?journalID=RSPB&manu=(Document%20not%20available)) which will take you to your unique entry in the Dryad repository. If you have already submitted your data to dryad you can make any necessary revisions to your dataset by following the above link. Please see <https://royalsocietypublishing.org/journals/ethics-policies/data-sharing-mining/> for more details.

6) For more information on our Licence to Publish, Open Access, Cover images and Media summaries, please visit <https://royalsocietypublishing.org/journals/authors/author-guidelines/>.

Sincerely,

Professor John Hutchinson, Editor
mailto: proceedingsb@royalsocietypublishing.org

Associate Editor
Comments to Author:

We thank the authors for taking the time to revise their article. Both referees are now happy with the manuscript. Please address the two very minor comments by referee one. Thank you for submitting your work to Proceedings B.

Reviewer(s)' Comments to Author:

Referee: 2

Comments to the Author(s).

I am satisfied with the authors' responses to the referee comments, as well as the revised manuscript. I think it is now ready for publication, pending two very minor points to be addressed:

1. The year for the Bicknell & Paterson reference should be 2018 (not 2017)
2. In the new section on appendage tagmatization (lines 229-234), a reference could be made to another recent study (Holmes et al. 2019, *Jour. Systematic Palaeo*) that suggests some trilobites show possible anterior-posterior differentiation of the biramous limbs - in this particular case, the

exopods of *Redlichia rex* (see their fig. 20), though Holmes et al. (2019) note that similar A-P trends occur in the appendages of other trilobite taxa such as *Eoredlichia intermediata* and *Triarthrus eatoni*. So it would seem A-P differentiation of the appendages in arthropods could be quite widespread across the clade, but expressed in different ways. Something to consider.

Referee: 1

Comments to the Author(s).

This new resubmission is more informative than the previous one and the comparative anatomy with other arthropods is insightful.

Author's Response to Decision Letter for (RSPB-2019-2371.R0)

See Appendix B.

Decision letter (RSPB-2019-2371.R1)

28-Oct-2019

Dear Dr Zhai

I am pleased to inform you that your manuscript entitled "Fine-scale appendage structure of the Cambrian trilobitomorph *Naraoia spinosa* and its ontogenetic and ecological implications" has been accepted for publication in *Proceedings B*.

Your article has been estimated as being 8 pages long. Our Production Office will be able to confirm the exact length at proof stage.

Open Access

You are invited to opt for Open Access, making your freely available to all as soon as it is ready for publication under a CC BY licence. Our article processing charge for Open Access is £1700. Corresponding authors from member institutions

Paper charges

Sincerely,

Editor, Proceedings B
mailto: proceedingsb@royalsociety.org

Appendix A

Responses to the reviewers' comments on our manuscript RSPB-2019-1721 "Fine-scale appendage structure of the Cambrian trilobitomorph *Naraoia spinosa* and its ontogenetic and ecological implications"

We greatly appreciate the valuable comments from the two reviewers and the associate editor. Most of these comments help improve the early version of our manuscript and were taken into careful consideration when we revised the manuscript. We address the comments as follows.

Comments from Associate Editor

Thank you for submitting your manuscript to Proceedings B. The reviewers have come back with rather contrasting reviews. Whilst Reviewer 2 is very happy with the manuscript and has very minor suggestions, Reviewer 1 would like to see considerable modifications. In order for the manuscript to appeal to the broad readership of PRSB, Reviewer 1 has suggested that the ontogeny of *Naraoia* be better framed within a broader comparison of the appendage anatomy of other taxa (e.g. *Tegopelte*, *Xandarella*, and trilobites such as agnostids). Additionally, Reviewer 1 has suggested the authors revisit the phylogenetic affinities of *Naraoia* in light of the recent phylogenetic analysis of Moysiuk and Caron (2019).

Response

Our study is almost unique in having access to ontogenetic data for appendages for Artiopoda. Not even trilobites have this information. This means that it isn't currently possible to formulate ontogenetic characters to shed light on naraoiid relationships because all other fossil terminals would be coded as question marks. We hope reviewer 1 will understand why our paper is fundamentally about ontogeny and functional morphology rather than about phylogeny. We attempted to add ontogenetic characters for the protopod (our novel data) to the Mayers et al (2018) naraoiid matrix (we picked it rather than the Moysiuk and Caron matrix because it has a comparable sampling of non-naraoiid artiopodans but it samples naraoiids densely, whereas the latter codes them as one genus-level terminal, *Naraoia*). However, no other artiopodans were codable for the characters. We hope this inspires others to work up comparable data.

Comparison with agnostids is definitely relevant in terms of documenting ontogenetic changes, since Müller and Walossek (1987) meticulously documented the appendages of juveniles and some coarser data for adults are now available (Moysiuk and Caron 2019). We stress that these sample different genera so they don't offer the precision that we have for a single species. Nonetheless, because agnostids are to some researchers relevant to artiopodan systematics, we added a section on their ontogeny, as follows:

"This suggests that this trend may be a plesiomorphic character shared by other artiopodans, but few comparable data are available to test its distribution. In the case of agnostids, a clade that has long been the subject of debate as to whether or not they are allied to trilobites (reviewed by Moysiuk and Caron [26]), comparison of juveniles [27] and adults suggests limited ontogenetic change [26], although comparison is only possible between different genera, and the adult morphology is known in much coarser detail than is available for *N. spinosa*."

The requested comparison with xandarellids has been added, with reference to the recently published microCT data for *Sinoburius* (Chen et al. 2019 [28]). We have made this a discussion about tagmatisation of the appendages.

Comments from Reviewer 1

The anatomical insights gained by computed microtomography are an excellent opportunity to produce new phylogenetic hypotheses. This paper needs a more thorough comparison with other trilobitomorphs pointing out specific trends (ecological or ontogenetic) found in *Naraoia* and other artiopodans. A phylogenetic analysis adding this new information into an existing matrix could allow making more sound inferences about the anatomy of other closely related arthropods, such as xandarellids. I would suggest complementing the discussion in this paper by exploring the evolutionary affinities of *Naraoia* by expanding the matrix published in Moysiuk and Caron (2019).

Moysiuk J, Caron J-B. 2019. Burgess Shale fossils shed light on the agnostid problem. *Proc. R. Soc. B* 286:20182314

Response

As explained above, our novel data are mostly relevant to ontogenetic characters, which cannot presently be coded for allied taxa. The Moysiuk and Caron analysis was designed to test the affinities of agnostids in the context of Arthropoda as a whole but it included just a single naraoiid terminal. As we note in our reply to the Associate Editor, we attempted to add new characters to the more densely sampled (for naraoiids) 2018 matrix by the same team but the codings were question marks in all terminals but *N. spinosa*, as ontogenetic data for appendages are so rare. We have taken on board the reviewer's suggestion to make additional ecological and ontogenetic comparisons with Cambrian artiopodans or putative artiopodans. Along these lines (and following advice from reviewer 2) we have added more text on the implications of antero-posterior differentiation of the protopods in terms of feeding mode, added a section on xandarellids to discuss tagmatisation of the appendages in Artiopoda, and have added ontogenetic comparisons with what is known for agnostids.

Comments from Reviewer 2

General comments

This is an excellent paper that describes the exquisitely preserved 3D morphology of juvenile and adult specimens of the Cambrian arthropod *Naraoia spinosa* from the Chengjiang Biota of China. It provides much needed new data on the morphology of this iconic group, and presents important novel information about the development and feeding modes of one of the oldest arthropods in the fossil record. Overall, the descriptions are accurate, the discussion is well written and argued, and the illustrations are of a high standard and show the necessary details. I think the manuscript could be published after very minor revision, provided the authors consider the points below:

1. It might be worth mentioning in the 'Materials' section that the specimens are partially preserved as pyrite and/or iron oxide. This preservation mode, with structures such as the appendages being

replicated by a much denser material than the surrounding matrix, is obviously why such specimens are conducive for micro-CT scanning and 3D imaging. Perhaps also citing a reference or two that discusses this preservation mode in Chengjiang fossils might be useful?

Response

We have added a passage to the Materials section as recommended.

Comments

2. I think there could be a bit more discussion on the functional differentiation of gnathobase morphology along the A-P axis of the adult, particularly in the context of other Cambrian and modern taxa. It is clear from the figures (especially Fig. 3) that the gnathobasic spines become more robust on the more posterior appendage pairs. This is a morphological trend seen in adult specimens of durophagous arthropods such as *Limulus polyphemus* and the Cambrian taxon *Sidneyia inexpectans* – see a recent discussion by Bicknell et al. (2018, *Arthropod Structure & Development*, 47: 12-24). This arrangement indicates that the posterior appendages are used for crushing harder food items before passing them anteriorly towards the mouth, where they are manipulated by the more delicate spines on the anteriormost pairs of gnathobases.

Response

We agree with this interpretation and so have added the following:

“As well, the adult of *N. spinosa* depicts a morphological trend of shorter but more robust spines on protopods of posterior appendages, consistent with feeding behaviour in horseshoe crabs and durophagous Cambrian arthropods such as *Sidneyia*, in which the posterior appendages break up prey that is passed forward to the more delicately spinose anterior appendages [21].”

Comments

3. This may be beyond the scope of the present study, but have the authors carefully examined specimens of *N. spinosa* in their collections that have digestive (gut) tracts preserved to see if there are shelly fragments contained within? Cololites provide ‘smoking gun’ evidence that the animal was indeed capable of durophagy (see the discussion by Bicknell & Paterson 2018, *Biological Reviews*, 93: 754-784), thus further supporting the claims made here. I should be clear that the manuscript does not need to provide this evidence, as the arguments about the feeding modes of juvenile and adult *N. spinosa* is already well supported by the new morphological information, but it would be great to see new data on Cambrian cololites if it is available. Just a suggestion.

Response

None of us has seen gut contents (apart from sediment) in *N. spinosa*. In case other readers have the same question, we added a sentence to make this point (citing the Bicknell and Paterson review), as follows:

“No shelly fragments are known from the gut of our scanned specimens or other material of *N. spinosa*, making durophagy less confident than is the case for, e.g., *Sidneyia* [22, 23].”

Appendix B

Final edits to our manuscript RSPB-2019-2371 following instructions from the editor and comments from the referees

We are very happy that our manuscript has been accepted for publication in Proceedings B. We appreciate the valuable comments from the two referees and the editor in this final round of review. In this version, we address the comments from the referees and go through the checklist provided by the editor (see below). Furthermore, we make a few minor corrections to the manuscript (detailed below; also highlighted in the manuscript).

Instructions from Editor

Response

Yes. The present document includes Response to Referees, as well as a copy of the manuscript text with "tracked changes" at its bottom.

Instruction

1) A text file of the manuscript (doc, txt, rtf or tex), including the references, tables (including captions) and figure captions. Please remove any tracked changes from the text before submission. PDF files are not an accepted format for the "Main Document".

Response

Yes. A text file "RSPB-2019-2371_clean version of manuscript text" has been uploaded, in which all track changes have been removed.

Instruction

2) A separate electronic file of each figure (tiff, EPS or print-quality PDF preferred). The format should be produced directly from original creation package, or original software format. PowerPoint files are not accepted.

Response

Yes. All figures are now uploaded as tiff and high-quality pdf formats.

Instruction

3) Electronic supplementary material: this should be contained in a separate file and where possible, all ESM should be combined into a single file. All supplementary materials accompanying an accepted article will be treated as in their final form. They will be published alongside the paper on the journal website and posted on the online figshare repository. Files on figshare will be made available approximately one week before the accompanying article so that the supplementary material can be attributed a unique DOI.

Response

Yes. The supplementary figures have been uploaded as a separate file.

Instruction

Response

Yes. The media summary is as follows:

Trilobites are among the most iconic groups of animals in the entire fossil record. While a substantial body of data documents how the mineralised exoskeleton changes through the course of development in trilobites, very little is known about how development affected the appendages. Here we use computed microtomography to reconstruct details of the appendages of a close relative of trilobites, *Naraoia spinosa*, from the early Cambrian (ca 518 million years ago) of China. CT scans of juvenile specimens compared to adults show that the base of the appendages is strikingly different between developmental stages, indicating differences in feeding strategy.

Instruction

NB. From April 1 2013, peer reviewed articles based on research funded wholly or partly by RCUK must include, if applicable, a statement on how the underlying research materials –

such as data, samples or models – can be accessed. This statement should be included in the data accessibility section.

[http://datadryad.org/submit?journalID=RSPB&manu=\(Document not available\)](http://datadryad.org/submit?journalID=RSPB&manu=(Document not available)) which will take you to your unique entry in the Dryad repository. If you have already submitted your data to dryad you can make any necessary revisions to your dataset by following the above link.

Please see <https://royalsociety.org/journals/ethics-policies/data-sharing-mining/> for more details.

Response

Yes. We have deposited our micro-CT data in Dryad. This is noted in “Data accessibility”. The title of our data in Dryad is “Computed Tomography data of the Cambrian euarthropod *Naraoia spinosa* from Chengjiang biota of China”. The doi of our data will be <https://doi.org/10.5061/dryad.jdfn2z372>.

Referees’ Comments

Referee: 2

Comments to the Author(s).

I am satisfied with the authors' responses to the referee comments, as well as the revised manuscript.

I think it is now ready for publication, pending two very minor points to be addressed:

1. The year for the Bicknell & Paterson reference should be 2018 (not 2017)

Response

Yes. This has been corrected. Please see reference 23 in the reference list (line 330).

Comment

2. In the new section on appendage tagmatization (lines 229-234), a reference could be made to another recent study (Holmes et al. 2019, Jour. Systematic Palaeo) that suggests some trilobites show possible anterior-posterior differentiation of the biramous limbs - in this particular case, the exopods of *Redlichia rex* (see their fig. 20), though Holmes et al. (2019) note that similar A-P trends occur in the appendages of other trilobite taxa such as *Eoredlichia intermediata* and *Triarthrus eatoni*. So it would seem A-P differentiation of the appendages in arthropodans could be quite widespread across the clade, but expressed in different ways. Something to consider.

Response

Yes. We have added a sentence to summarize the findings by Holmes et al. (2019) (as ref. 28; former references 28, 29 and 30 have been re-numbered). Please see lines 231–235, reference numbers in the text, as well as the reference list.

An additional minor change:

In line 46, we deleted "Late" from " Přidolian (Late Silurian)". It should just say " Přidolian (Silurian)". The Silurian doesn't formally have an Early and Late subdivision in the current scheme.